# Acute and Subchronic (28-day) Oral Toxicity Studies on the Film Formulation of k-Carrageenan and Konjac Glucomannan for Soft Capsule Application

**Ni Nyoman Wiwik Sutrisni [1], Sundani Nurono Soewandhi [1], I Ketut Adnyana [2] and Lucy D N Sasongko [1,*]**

[1] Department of Pharmaceutics, School of Pharmacy, Institut Teknologi Bandung, Bandung 40132, Indonesia; wiwik.sutrisni@gmail.com (N.N.W.S.); sundani@fa.itb.ac.id (S.N.S.)

[2] Department of Pharmacology and Clinical Pharmacy, School of Pharmacy, Institut Teknologi Bandung, Bandung 40132, Indonesia; ketut@fa.itb.ac.id

[*] Correspondence: lucys@itb.ac.id; Tel.: +62-22-250-4852

**Abstract:** The aim of this study was to investigate the acute and subchronic toxicity of a film formulation that combines κ-Carrageenan and konjac glucomannan for soft capsule application. For the acute toxicity study, a dose of 2000 mg/kg body weight (bw) of the film suspension was administered orally to rats. The animals were observed for toxic symptoms and mortality daily for 14 days. In a subchronic toxicity study, the film suspension, at doses of 10, 30 and 75 mg/kg bw for 28 days, were orally administered to rats. After 28 days, the rats were sacrificed for hematological, biochemical and histological examination. In the acute toxicity study, neither signs of toxicity nor death among the rats were observed for up to 14 days of the experimental period. The results of the subchronic toxicity study show that there were no significant changes observed in the hematology and organ histology. Some alterations to the relative organ weight and blood biochemistry were observed, but they were considered to be temporary effects and not an indication of toxic effects. The overall findings of this study indicate that the film formulation of κ-Carrageenan and konjac glucomannan is non-toxic up to a dose of 75 mg/kg bw, which could be considered a safe dose for soft capsule application.

**Keywords:** κ-Carrageenan; Konjac Glucomannan; soft capsule; toxicity

## 1. Introduction

Capsules are an alternative to tablets for the oral delivery of therapeutic compounds. Capsule shells, available as hard or soft shells, are formulated from gelatin or a non-gelatin polymeric material. Market analysis of encapsulated medications and biologically active dietary supplements indicates that attention need to be paid to the search for alternatives to the traditionally used gelatin. This factor determines the relevance of the development of technology for the production of capsules from hydrocolloids of plant origin [1,2]. A combination of κ-Carrageenan, an anionic sulfated linear polysaccharide extracted from red seaweed, and konjac glucomannan, a natural hydrocolloid gum derived from the tubers of *Amorphophallus konjac* K. Koch, could potentially be used in soft capsule formulation due to their synergic interaction. The addition of konjac glucomannan to κ-carrageenan based gels is often employed in food applications to decrease syneresis, reduce brittleness and increase the linear regime elastic modulus [3]. The ratio of konjac glucomannan to κ-carrageenan is responsible for the texture and rheological properties of the system. An excess of konjac glucomannan reinforces the elastic network and increases the fracture strain and stress considerably [3,4]. Carrageenan has been associated with the induction and promotion of intestinal

neoplasms and ulceration in numerous animal experiments. Nevertheless, based on short- and long-term studies of toxicology reviewed by the Joint FAO/WHO Expert Committee on Food Additives (JECFA), it was concluded that carrageenan is safe to consume, although the recommended Acceptable Daily Intake is "not specified" for carrageenan [5]. The toxicology of carrageenan is related to the unique chemical structure of carrageenan. It can be categorized as "undegraded" or high molecular weight carrageenan and "degraded" or low molecular weight carrageenan, which is associated with an inflammatory effect. The degradation of carrageenan through depolymerization may be due to acidic exposure. The cytotoxic study on κ-carrageenan concluded that degraded κ-carrageenan inhibited cell proliferation in four types of cell line. However, no cytotoxic effect was found in undegraded carrageenan [6]. To determine the safety of κ-carrageenan film with konjac glucomannan formulation for soft capsule application, film suspension was administered to rats for oral acute and subchronic toxicity studies. The subchronic toxicity study was conducted for 28 days, during which the film suspension was orally administered, and after 28 days toxicity parameters such as body weight changes, relative organ weight, hematology and clinical biochemistry profiles, and organ histopathology were examined.

## 2. Materials and Methods

### 2.1. Test Substance Preparation

The test substance was a film formulation of κ-carrageenan added konjac glucomannan for soft capsule shell application. The formula was as follows: κ-carrageenan 4%, konjac glucomannan 1%, sodium carbonate 0.002%, glycerin 12%, sorbitol solution 7.5%, methyl paraben 0.18%, sorbic acid 0.1% and balance purified water ad 100%. κ-carrageenan was purchased from Karaindo Indonesia (RVN K2), Konjac Glucomannan was purchased from Shannxi Jiahe, Sodium carbonate was purchased from Merck, Glycerin was purchased from PT Sumi asih, Sorbitol solution 70% was purchased from Cargill, Methylparaben was purchased from UENO, and Sorbic acid was purchased from Merck.

The film was prepared by melting the polymers (κ-carrageenan and konjac glucomannan) in water solution at 80–90 °C and mixing in other ingredients. The films were molded manually in a petri glass with a diameter of 9–10 mm, and 15 g of mass to provide a film thickness of 0.7–0.9 mm, then they were aged at an ambient temperature for 3 days to obtain the film, with loss on drying within a range of 25–30%.

### 2.2. Experimental Animals

Normal female Wistar rats were used for acute toxicity study. Normal male and female Wistar rats were used for the subchronic (28-day) study. The animals were 6–8 weeks old, and were stabilized for seven days prior to the experiments. They were acclimatized at room temperature, with 12 h light and 12 h dark cycle, as well as free access to a standard pellet diet and water ad libitum. During acclimatization, the rats were randomized into experimental and control groups. All experimental procedures were in compliance with the Guide for the Care and Use of Laboratory Animals and approved by the local animal care committee, The Health Research Ethic Committee of Faculty of Medicine Universitas Padjajaran Bandung with ethical approval no 418/UN6.C1.3.2/KEPK/PN/2016.

### 2.3. Acute Oral Toxicity Study

An acute oral toxicity study was performed according to the Organization of Economic Co-operation and Development (OECD) guideline 420 for testing a chemical [7]. Five female rats were used. A test suspension of 20% (200 mg/mL) in purified water was orally administered only once at a single dose of 2000 mg/kg bw, whereas the control group only received 2 mL of purified water as a vehicle. All rats were then granted free access to food and water, observed for any signs of acute toxicity for 24 h with special attention to the first 4 h and once daily for 14 days.

*2.4. Subchronic Oral Toxicity Study*

A subchronic oral toxicity study was preformed according to the Organization of Economic Co-operation and Development (OECD) guideline 407 for testing a chemical [8]. The animals were randomly divided as follows: Group 1 (control) and Group 4 (high dose) contained 10 animals/sex/group, and Group 2 (low dose) and Group 3 (middle dose) contained 5 animals/sex/group. At the end of the 28-day treatment period, 5 animals/sex/group were necropsied. The remaining 5 animals/sex/group in Group 1 and Group 4 remained untreated for up to 14 days (satellite group). At the end of the recovery period, they were necropsied. All animals were fasted overnight prior to necropsy and euthanized by carbon dioxide inhalation. Relative organ weights, hematological and biochemical profiles were determined.

The target doses for Groups 1–4 were 0, 10, 30, and 75 mg film/kg bw/day, respectively. The dose levels were selected based on the dose conversion in soft capsule application. The film will be applied for human use as an 8-oblong soft capsule with a 250 mg weight/capsule/day dosage, which is equivalent to 4.166 mg/kg bw. The middle dose was converted by the following calculation:

$$7 \times 4.166 \frac{mg}{kg} = 29.16 \frac{mg}{kg} bw \sim 30 \frac{mg}{kg} bw$$

The low and high doses were set as 1/3- and 2.5-fold of the middle dose, respectively. These doses were selected based on the dose level guideline of OECD 407, which states that 2–4-fold intervals are frequently optimal for setting the descending dose levels.

The test substances were prepared as the water suspension of chopped film at a concentration of 2, 6 and 15 mg/mL, for the low (10 mg/kg bw), middle (30 mg/kg bw) and high doses (75 mg/kg bw), respectively, to deliver not more than 2 mL of it to each testing animal. Visual observation of the test substances indicated that the water suspension of chopped film was stable for 7 days at room temperature without changes viscosity of suspension.

2.4.1. Body Weight and Clinical Observation

The body weight of each rat was weighed before the test, weekly during the study and on the day of sacrifice. All animals were observed daily for mortality and their general condition.

2.4.2. Relative Organ Weight Determination

The organs, namely the heart, liver, spleen, lung, kidney, adrenal, testes, seminal vesicles, ovarium, and uterus, were removed from the rats and weighed. The relative organ weight was calculated as follows:

Relative organ weight = (organ weight (g) / body weight of the animal on sacrifice day (g)) $\times$ 100

2.4.3. Hematological Analysis

The following hematological parameters were evaluated on a Medonic M-Series Hematology Analyzer: white blood cell counts, hemoglobin, red blood cell counts, mean corpuscular volume (MCV), mean corpuscular hemoglobin (MCH), mean corpuscular hemoglobin concentration (MCHC), and hematocrit and platelet counts [8].

2.4.4. Biochemical Analysis

Blood samples were collected from a cardiac puncture in a non-heparinized tube, and then centrifuged at 10,000 rpm for 5 min. A serum was separated and analyzed for biochemical parameters on a Photometer 5010 V5+ based on an enzymatic colorimetric test, including blood glucose, cholesterol, triglyceride, aspartate aminotransferase (AST), alanine aminotransferase (ALT), blood urea and blood creatinine. The blood glucose determination was performed according to the method of Barham and Trinder [9]. The total cholesterol was determined by the CHOD-PAP

(Cholesterol oxidase/p-aminophenazone) method [10]. Triglyceride was determined by the GPO-PAP (Glycerol-3-phosphate oxidase/p-aminophenazone) method [11]. Aspartate Amino Transferase (AST) and Alanine Amino Transferase (ALT) were determined by the kinetic method for the determination of AST and ALT activities according to the recommendation of the expert panel of the International Federation of Clinical Chemistry (IFCC) [12,13]. Creatinine was determined by the Jaffe reaction, based on a colorimetric test for kinetic determination of creatinine at 25 °C and 37 °C without deproteinization [14]. Blood urea was determined by the Berthelot method.

### 2.4.5. Histopathology Study

The organs, namely, the liver, heart, spleen, lung and kidney, were carefully excised and preserved in a fixation medium of 10% buffered formalin for histopathological study. The organ paraffin section was prepared, stained with hemotoxylin and eosin and processed for a light microscope fitted with an optilab viewer [15]. The histopathology study was conducted at the Histopathology Laboratory, The department of pathology and anatomy, Faculty of Medicines, Universitas Padjajaran, Bandung.

### *2.5. Statistical Analysis*

Data were expressed as mean $\pm$ standard deviation (SD) and analyzed statistically by a one-way Analysis of Variance (ANOVA). Then, Tukey's post hoc test compared the results with the control group using statistical software SPSS version 23, and $p < 0.05$ was considered to be different to a statistically significant degree.

## 3. Results

### *3.1. Acute Oral Toxicity Study*

In all five female testing animals, signs of neither toxicity nor death among the rats were observed during the 14 days of the acute toxicity experimental period, after the administration of a single oral dose of the testing suspension of 2000 mg/kg bw. The body weight measurement resulted in an average body weight of 193.20 $\pm$ 7.43, 204.20 $\pm$ 8.29, and 211.00 $\pm$ 8.83 g for Days 1, 7 and 14, respectively. The body weight gradually increased within normal range of body weight gain. After 14 days of treatment, all tested animals were subjected to gross necropsy with no abnormality of organs observed in the macroscopic observation. This result suggests that the film formulation of κ-carrageenan with konjac glucomannan was not toxic, after an acute exposure.

### *3.2. Subchronic 28-day Oral Toxicity Study*

#### 3.2.1. Body Weight and Clinical Observation

Neither deaths nor obvious clinical signs of toxicity in the rats were observed for all groups, including the group that received the highest tested dose of 75 mg/kg bw.

#### 3.2.2. Relative Organ Weight Determination

Tables 1 and 2 show the relative organ weights of the male and female rats after 28 days of administration of the film suspension. The relative organ weight of each organ evaluated and calculated at necropsy in the treatment groups did not show a significant difference ($p > 0.05$) compared to the control, except for the seminal vesicles at a dose of 75 mg/kg bw for male rats and the liver at a dose of 75 mg/kg bw for female rats. A significant reduction of the relative weight of the seminal vesicles at a high dose of the test substance was observed. The relative liver weight of female rats in the 75 mg/kg bw group increased compared to the control group ($p < 0.05$).

**Table 1.** Relative organ weights of male rats treated with different doses of the film formulation for 28 days.

| Organs | Groups | | | | | |
|---|---|---|---|---|---|---|
| | **Control** | **Satellite Control** | **10 mg/kg** | **30 mg/kg** | **75 mg/kg** | **Satellite 75 mg/kg** |
| Heart | 0.39 ± 0.04 | 0.36 ± 0.03 | 0.36 ± 0.04 | 0.43 ± 0.06 | 0.36 ± 0.04 | 0.34 ± 0.03 |
| Liver | 3.00 ± 0.28 | 3.32 ± 0.17 | 2.72 ± 0.24 | 2.98 ± 0.26 | 3.28 ± 0.34 | 3.22 ± 0.18 |
| Spleen | 0.24 ± 0.02 | 0.30 ± 0.06 | 0.20 ± 0.03 | 0.24 ± 0.03 | 0.24 ± 0.10 | 0.26 ± 0.07 |
| Kidneys | 0.68 ± 0.05 | 0.70 ± 0.09 | 0.66 ± 0.05 | 0.70 ± 0.03 | 0.69 ± 0.07 | 0.66 ± 0.05 |
| Lungs | 0.69 ± 0.18 | 0.67 ± 0.13 | 0.53 ± 0.11 | 0.60 ± 0.11 | 0.74 ± 0.15 | 0.71 ± 0.08 |
| Adrenal | 0.02 ± 0.009 | 0.02 ± 0.004 | 0.03 ± 0.011 | 0.02 ± 0.008 | 0.03 ± 0.005 | 0.02 ± 0.003 |
| Testicles | 1.05 ± 0.10 | 1.17 ± 0.08 | 1.03 ± 0.11 | 1.10 ± 0.05 | 1.25 ± 0.22 | 1.04 ± 0.09 |
| Seminal vesicles | 0.50 ± 0.10 | 0.35 ± 0.06 | 0.46 ± 0.06 | 0.44 ± 0.09 | 0.28 ± 0.04 * | 0.32 ± 0.05 |

Values are expressed as the mean ± SD; * significantly different from the control group ($p < 0.05$).

**Table 2.** Relative organ weights of female rats treated with different doses of the film formulation for 28 days.

| Organs | Groups | | | | | |
|---|---|---|---|---|---|---|
| | **Control** | **Satellite Control** | **10 mg/kg** | **30 mg/kg** | **75 mg/kg** | **Satellite 75 mg/kg** |
| Heart | 0.40 ± 0.04 | 0.36 ± 0.05 | 0.42 ± 0.06 | 0.40 ± 0.06 | 0.38 ± 0.05 | 0.38 ± 0.03 |
| Liver | 3.25 ± 0.42 | 3.43 ± 0.34 | 2.91 ± 0.27 | 2.76 ± 0.27 | 3.91 ± 0.15 * | 3.42 ± 0.31 |
| Spleen | 0.39 ± 0.14 | 0.35 ± 0.09 | 0.38 ± 0.06 | 0.44 ± 0.09 | 0.42 ± 0.16 | 0.49 ± 0.03 |
| Kidneys | 0.66 ± 0.05 | 0.68 ± 0.05 | 0.67 ± 0.04 | 0.60 ± 0.17 | 0.67 ± 0.05 | 0.64 ± 0.07 |
| Lungs | 0.74 ± 0.18 | 0.72 ± 0.13 | 0.90 ± 0.37 | 1.02 ± 0.53 | 0.83 ± 0.07 | 0.85 ± 0.14 |
| Adrenal | 0.04 ± 0.010 | 0.04 ± 0.002 | 0.03 ± 0.008 | 0.04 ± 0.012 | 0.04 ± 0.005 | 0.04 ± 0.004 |
| Ovaries | 0.06 ± 0.015 | 0.06 ± 0.014 | 0.07 ± 0.014 | 0.05 ± 0.017 | 0.07 ± 0.007 | 0.07 ± 0.006 |
| Uterus | 0.22 ± 0.03 | 0.24 ± 0.10 | 0.23 ± 0.07 | 0.20 ± 0.03 | 0.21 ± 0.07 | 0.26 ± 0.09 |

Values are expressed as the mean ± SD; * significantly different from the control group ($p < 0.05$).

### 3.2.3. Hematological Analysis

Tables 3 and 4 show the hematological profile of male and female rats, after 28 days of the administration of the film suspension. The oral administration of the suspension of the film formulation produced no effect on most hematological parameters. The result show that none of the groups differed significantly when compared to the control for all parameters.

**Table 3.** Hematological profile of male rats treated with different doses of the film formulation for 28 days.

| Parameter | Unit | Groups | | | | | |
|---|---|---|---|---|---|---|---|
| | | **Control** | **Satellite Control** | **10 mg/kg** | **30 mg/kg** | **75 mg/kg** | **Satellite 75 mg/kg** |
| White blood cell counts | $10^3/mm^3$ | 8.72 ± 1.78 | 8.90 ± 2.34 | 5.60 ± 1.02 | 8.72 ± 3.01 | 6.00 ± 1.65 | 10.56 ± 2.33 |
| Hemoglobin (Hb) | g/dL | 8.40 ± 0.75 | 8.90 ± 1.32 | 8.16 ± 0.54 | 8.72 ± 0.59 | 9.04 ± 0.96 | 9.12 ± 0.59 |
| Red blood cell counts | $10^6/mm^3$ | 5.06 ± 0.93 | 4.62 ± 0.63 | 5.41 ± 0.73 | 6.48 ± 1.39 | 4.53 ± 0.60 | 5.07 ± 0.44 |
| Mean corpuscular volume (MCV) | fL | 37.04 ± 0.60 | 37.20 ± 0.41 | 37.27 ± 0.21 | 37.33 ± 0.58 | 36.42 ± 0.49 | 37.22 ± 0.47 |
| Mean corpuscular hemoglobin (MCH) | pg/sel | 16.95 ± 2.55 | 19.28 ± 1.18 | 15.25 ± 1.83 | 14.08 ± 3.78 | 20.02 ± 0.63 | 18.03 ± 1.06 |
| Mean corpuscular hemoglobin concentration (MCHC) | g/dL | 45.79 ± 7.11 | 51.85 ± 3.56 | 40.91 ± 4.85 | 37.67 ± 9.70 | 54.98 ± 1.87 | 48.47 ± 3.32 |

**Table 3.** *Cont.*

| Parameter | Unit | Groups | | | | | |
|---|---|---|---|---|---|---|---|
| | | Control | Satellite Control | 10 mg/kg | 30 mg/kg | 75 mg/kg | Satellite 75 mg/kg |
| Hematocrit | % | 37.44 ± 6.85 | 34.40 ± 4.98 | 40.32 ± 5.47 | 48.32 ± 10.21 | 32.96 ± 4.17 | 37.76 ± 3.32 |
| Platelet | $10^3$/mL | 536.00 ± 106.73 | 509.25 ± 74.00 | 519.60 ± 43.64 | 556.33 ± 135.88 | 609.80 ± 81.23 | 543.80 ± 58.26 |

Values are expressed as the mean ± SD; * significantly different from the control group ($p < 0.05$).

**Table 4.** Hematological profile of female rats treated with different doses of the film formulation for 28 days.

| Parameter | Unit | Groups | | | | | |
|---|---|---|---|---|---|---|---|
| | | Control | Satellite Control | 10 mg/kg | 30 mg/kg | 75 mg/kg | Satellite 75 mg/kg |
| White blood cell counts | $10^3$/mm$^3$ | 9.68 ± 2.46 | 7.30 ± 1.89 | 12.24 ± 4.54 | 10.72 ± 3.81 | 6.00 ± 1.17 | 6.88 ± 2.67 |
| Hemoglobin (Hb) | g/dL | 7.36 ± 1.08 | 8.90 ± 0.38 | 8.40 ± 0.57 | 8.40 ± 0.63 | 7.84 ± 1.08 | 8.48 ± 0.52 |
| Red blood cell counts | $10^6$/mm$^3$ | 3.89 ± 0.77 | 5.28 ± 0.81 | 4.69 ± 0.54 | 4.13 ± 0.64 | 5.17 ± 0.64 | 5.62 ± 1.37 |
| Mean corpuscular volume (MCV) | fL | 36.62 ± 0.80 | 37.89 ± 0.70 | 37.20 ± 0.75 | 37.20 ± 0.33 | 37.44 ± 0.69 | 36.34 ± 1.15 |
| Mean corpuscular hemoglobin (MCH) | pg/sel | 19.41 ± 3.65 | 17.10 ± 2.15 | 18.05 ± 1.67 | 20.56 ± 1.87 | 15.47 ± 3.39 | 15.78 ± 3.55 |
| Mean corpuscular hemoglobin concentration (MCHC) | g/dL | 53.08 ± 10.33 | 45.12 ± 5.41 | 48.49 ± 4.08 | 55.29 ± 5.31 | 41.33 ± 9.14 | 43.25 ± 8.65 |
| Hematocrit | % | 28.48 ± 5.81 | 40.00 ± 6.13 | 34.88 ± 4.14 | 30.72 ± 4.85 | 38.72 ± 5.11 | 40.64 ± 9.09 |
| Platelet | $10^3$/mL | 548.75 ± 63.62 | 397.75 ± 66.53 | 551.20 ± 56.92 | 591.20 ± 68.43 | 492.00 ± 31.77 | 413.67 ± 90.84 |

Values are expressed as the mean ± SD; * significantly different from the control group ($p < 0.05$).

### 3.2.4. Biochemical Analysis

Tables 5 and 6 show the biochemical profile of male and female rats after 28 days of the administration of the film suspension. Some biochemical parameters were affected by the oral administration of the suspension of the film formulation. In the male rats treated with 10 mg/kg bw, the concentration of glucose was significantly lower than that of the control group. In the male rats treated with 30 mg/kg bw, aspartate amino transferase (AST) was significantly increased. In the male rats treated with 75 mg/kg bw, the concentration of triglyceride was significantly increased. In the female rats treated with 30 mg/kg bw, the urea and triglyceride concentrations were significantly lower than those of the control group. However, in the group treated with 75 mg/kg bw, the concentration of triglyceride was significantly increased ($p < 0.05$).

**Table 5.** Biochemical profile of male rats treated with different doses of the film formulation for 28 days.

| Parameter | Unit | Groups | | | | | |
|---|---|---|---|---|---|---|---|
| | | Control | Satellite Control | 10 mg/kg | 30 mg/kg | 75 mg/kg | Satellite 75 mg/kg |
| Glucose | mg/dL | 117.80 ± 34.30 | 168.25 ± 19.53 | 77.00 ± 18.21 * | 85.00 ± 13.02 | 109.00 ± 9.03 | 141.40 ± 16.56 |
| Cholesterol | mg/dL | 95.40 ± 13.43 | 88.75 ± 4.03 | 87.00 ± 5.48 | 77.60 ± 9.50 | 71.40 ± 13.05 | 87.20 ± 6.46 |
| Triglyceride | mg/dL | 40.80 ± 9.78 | 53.55 ± 3.12 | 48.60 ± 3.29 | 51.00 ± 5.61 | 82.60 ± 19.88 * | 76.60 ± 8.79 * |
| Aspartate amino transferase (AST) | U/L | 209.60 ± 30.88 | 259.75 ± 13.38 | 231.20 ± 30.74 | 269.60 ± 7.93 * | 238.20 ± 18.75 | 258.60 ± 1.95 |
| Alanine amino transferaase (ALT) | U/L | 105.40 ± 27.66 | 113.50 ± 11.03 | 106.80 ± 15.43 | 118.60 ± 13.69 | 122.20 ± 10.71 | 111.80 ± 16.04 |
| Urea | mg/dL | 46.24 ± 13.16 | 53.55 ± 3.12 | 42.76 ± 2.92 | 51.40 ± 10.69 | 45.56 ± 10.72 | 48.86 ± 1.41 |
| Creatinine | mg/dL | 0.90 ± 0.21 | 0.51 ± 0.12 | 0.59 ± 0.24 | 0.61 ± 0.25 | 1.00 ± 0.11 | 0.53 ± 0.15 |

Values are expressed as the mean ± SD; * significantly different from the control group (p < 0.05).

**Table 6.** Biochemical profile of female rats treated with different doses of the film formulation for 28 days.

| Parameter | Unit | Groups | | | | | |
|---|---|---|---|---|---|---|---|
| | | Control | Satellite Control | 10 mg/kg | 30 mg/kg | 75 mg/kg | Satellite 75 mg/kg |
| Glucose | mg/dL | 118.20 ± 14.52 | 119.25 ± 27.69 | 96.40 ± 24.15 | 105.80 ± 25.64 | 161.40 ± 26.66 | 107.20 ± 17.24 |
| Cholesterol | mg/dL | 122.40 ± 29.71 | 94.50 ± 15.33 | 99.20 ± 29.39 | 109.80 ± 18.75 | 96.60 ± 15.14 | 92.00 ± 8.12 |
| Triglyceride | mg/dL | 53.80 ± 6.02 | 40.25 ± 4.57 | 44.40 ± 4.22 | 30.40 ± 6.19 * | 69.80 ± 9.96 * | 40.40 ± 12.44 |
| Aspartate amino transferase (AST) | U/L | 159.20 ± 9.09 | 293.00 ± 23.92 | 195.80 ± 38.62 | 205.60 ± 31.37 | 242.40 ± 16.80 * | 278.00 ± 22.67 |
| Alanine amino transferaase (ALT) | U/L | 104.60 ± 17.26 | 93.00 ± 23.40 | 97.00 ± 12.27 | 87.20 ± 10.64 | 120.80 ± 25.06 | 81.40 ± 21.22 |
| Urea | mg/dL | 48.24 ± 5.22 | 39.80 ± 6.59 | 44.06 ± 9.15 | 35.86 ± 2.77 * | 46.82 ± 6.40 | 42.12 ± 4.97 |
| Creatinine | mg/dL | 0.42 ± 0.18 | 0.41 ± 0.04 | 0.57 ± 0.18 | 0.71 ± 0.22 | 1.12 ± 0.10 * | 0.42 ± 0.06 |

Values are expressed as the mean ± SD; * significantly different from the control group ($p < 0.05$).

### 3.2.5. Histopathology Study

Figures 1 and 2 show the histopathology profile of male and female rats after 28 days of the administration of the film suspension. Light microscopic examination of the section of various organs, namely, the kidney, liver, heart, spleen and lung of rats from the control group and those treated with a low dose (10 mg/kg bw), medium dose (30 mg/kg bw) and high dose (75 mg/kg bw), showed a normal histology.

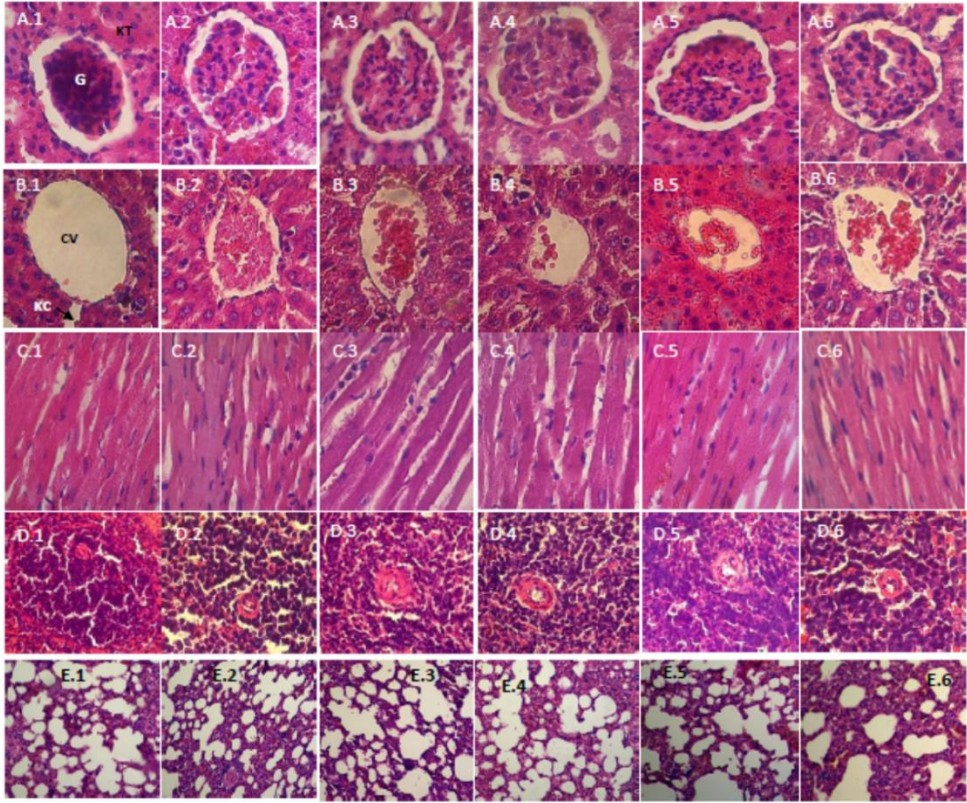

**Figure 1.** Histopathological examination (HE, 400×) of various organs of male rats in a subchronic oral toxicity study: (**A–E**) the kidney, liver, heart, spleen and lung, respectively; and (1–6) the control, satellite-control, the low, middle, and high dose groups, and the satellite-highest dose group (75 mg/kg bw), respectively. G, Glomerulus; KT, Kidney Tubules; CV, Central Vein; KC, Kupffer Cell.

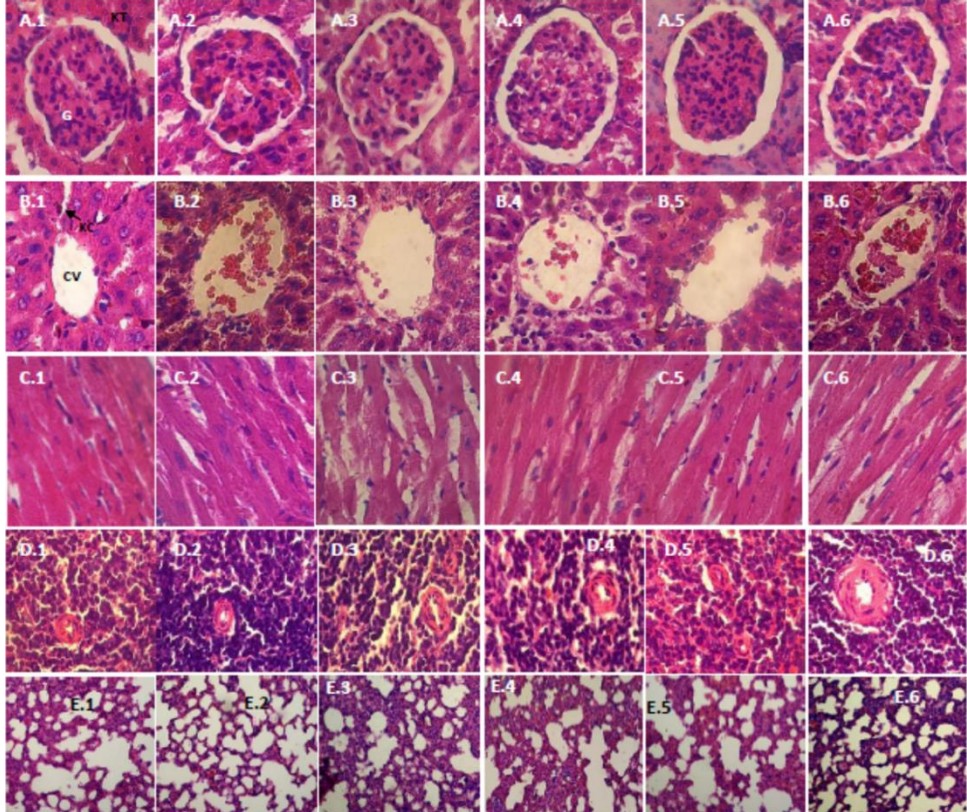

**Figure 2.** Histopathological examination (HE, 400×) of various organs of female rats in a subchronic oral toxicity study: (**A–E**) the kidney, liver, heart, spleen and lung, respectively; and (1–6) the control, satellite-control, the low, middle, and high dose groups, and the satellite-highest dose group (75 mg/kg bw), respectively. G, Glomerulus; KT, Kidney Tubules; CV, Central Vein; KC, Kupffer Cell.

## 4. Discussion

Carrageenan has been widely used in food, pharmaceutical and industrial applications. While carrageenan is generally recognized as safe (GRAS) by the United States Food and Drug Administration, the safety of ingested carrageenan remains questionable, as it may be altered by bacteria in the intestine, if the stomach has an acidic condition [6]. Konjac glucomannan has been recognized as GRAS by a consensus of scientific opinion since 1994 and it is regarded as a non-caloric food, which in its traditional form is a source of indigestible high-quality dietary fiber [4]. The present study aimed to investigate the safety of a combination of κ-carrageenan and konjac glucomannan in the formulation of film for soft capsule application. The film formulation contained 4% of κ-carrageenan and 1% of konjac glucomannan. The low polymer content was due to the limitation of high viscosity produced by the polymers, but this formulation could obtain an elastic film suitable for soft capsule application because the addition of small portion of konjac glucomannan to κ-carrageenan gel reduced the brittleness and increased the linear regime elastic modulus [3,4]. High water content of film was needed to maintain the elasticity of the film for encapsulation purpose. Doses of test substance for toxicity studies were confirmed by preparing the film suspension using calculated anhydrate film weight. As an initial step, an oral acute toxicity study was conducted, and the results show that the oral administration of film suspension was well tolerated. After a dose of 2000 mg/kg bw of the film suspension, administered to the rats, no acute toxicities, mortality or abnormal behavior was observed for up to 14 days. The results reveal that, according to the labeling and classification of acute systemic toxicity recommended by the OECD, the film formulation was assigned a Class 5 status, i.e., relatively low acute toxicity hazard with the oral $LD_{50}$ in the range of 2000–5000 mg/kg [7]. Further investigation was conducted to evaluate

the subchronic toxicity of the film formulation for up 28 days in rats to prepare safety data on this potential formulation.

The subchronic 28-day oral toxicity study of the film formulation demonstrated no adverse clinical signs or negative influences on behavior and mortality in the treatment groups. Analysis of the hematological parameters is important in assessing the toxic effects of test substances, as well as in determining the physiological and pathological status of the body, as variations in these parameters may indicate toxicity associated with the test substances and various diseases and conditions, including anemia, leukemia, reactions to inflammation and infections [16,17]. There was no significant difference in white blood cell counts, hemoglobin, red blood cell counts, MCV, MCH, MCHC, hematocrit and platelets between the treated groups and the control group, indicating that the film suspension had no effect on the circulating blood cells of the tested animals.

Relative organ weight profiles are essential and widely used for interpreting the effect of a test substance in a toxicity study. Alterations of the liver weight may suggest treatment-related changes including hepatocellular hypertrophy (e.g., enzyme induction or peroxisome proliferation), an elevated heart weight, which may be the only evidence of myocardial hypertrophy, changes in the kidney weight, which may reflect renal toxicity, and a variation in the adrenal gland weight, which may indicate hypertrophy, hyperplasia or atrophy associated with the test substance effects [18]. In the male groups, no significant changes in relative organ weight were observed between the treatment groups and the control group except in the seminal vesicle weight for the 75 mg/kg bw group. On the other hand, the 75 mg/kg bw satellite group with a two-week discontinuation of the test substance showed no significant changes, which might indicate that a reduction of the relative seminal vesicle weight might represent secondary effects of the treatment on the reproductive cycle rather than a direct toxic effect of the test substance [18]. In the female groups, only the relative liver weight in the 75 mg/kg bw group differed significantly, compared to the control group. The liver is the main organ for the metabolism and detoxification of drugs and environmental chemicals. Serum ALT activity has been historically used as a major biomarker for liver injury in preclinical studies. Damaged hepatocytes release ALT into the extracellular space, with a subsequent passage into the blood [19,20]. In this study, the serum ALT of the altered result showed normal values with respect to the control group. The ALT level (120.80 U/L) did not satisfy the diagnostic criteria for liver damage, compared to the control group (104.60 U/L). Liver histopathology examination of the female group in the 75 mg/kg bw did not show any alteration in the cell structure, inflammation, or any other unfavorable effects.

Hepatotoxicity can be characterized into two main groups, each with a different mechanism of injury: hepatocellular and cholestatic. Hepatocellular or cytolytic injury involves predominantly initial serum aminotransferase level elevations, usually preceding increases in total bilirubin levels and modest increases in alkaline phosphatase levels. Cholestatic injury is predominantly characterized by initial alkaline phosphatase level elevations that precede or are relatively more prominent than increases in the levels of serum aminotransferases [21]. In this study, a higher level of serum AST was observed in the 30 mg/kg bw male group and 75 mg/kg bw female group, but through analysis of the histopathological examination results, no significant damage was found to be induced by the administration of the film suspension.

The effect of the film suspension on the kidney was investigated by measuring the serum level of urea and creatinine, which are specific indicators of kidney function. An incremental change in these parameters indicates damage to the functional nephrons [22,23]. No significant changes were observed in the serum level of urea and creatinine except for the lower urea level in the male group at a dose of 30 mg/kg bw, but this was considered a normal value and was not dose-dependent. These findings were confirmed by the kidney relative weight and the histopathological observations of the kidney tissues, which indicated that the film suspension was not toxic to the kidneys.

The lipid profile is used to diagnose primary and secondary hyperliproteinemia, triglyceridemia, liver obstruction, and fatty liver disease [24]. High triglyceride and cholesterol levels are linked to atherosclerosis and predispose an individual to cardiovascular disease [25]. In this study, quantitative

measurements were conducted on the serum level of cholesterol and triglyceride. The results show that, in both the male and female groups, the film suspension administration reduced the serum level of cholesterol but did not differ significantly, compared to the control group. On the other hand, the triglyceride level tended to increase and was significantly higher at a dose of 75 mg/kg bw. Konjac glucomannan, as one component in the film formulation, has been touted for its potential beneficial effects in prolonging the gastric emptying time, decreasing the ingestion of foods that increase cholesterol and glucose concentrations, reducing the postprandial rise in plasma glucose, suppressing hepatic cholesterol synthesis, and increasing the fecal elimination of cholesterol that contains bile acids. Ah meta-analysis shows that glucomannan has an ability to lower triglycerides but its ability to preferentially lower triglycerides compared with other soluble fibers is not known. This may be related to its higher viscosity and thus its greater ability to alter the metabolic pathways of hepatic cholesterol and lipoprotein metabolism [26]. The opposite result obtained in this present study might be caused by the low concentration of glucomannan in the formulation. The serum triglyceride concentration, after the administration of the film suspension within the normal range, and the increasing effect of triglycerides was considered temporary, because a lower value was measured in the satellite groups when the discontinuation of the film suspension was conducted for two weeks.

The serum glucose concentration reflects the intestinal glucose absorption, hepatic glucose production, and tissue uptake of glucose [27]. After 28 days of the administration of the film suspension, the glucose level reduced, with the lowest value occurring at a dose of 10 mg/kg bw. As a hydrophilic polysaccharide, konjac glucomannan exhibits a highwater absorbency, and 1 g of material can absorb 50–100 g of water [28]. The high water-binding capacity and swelling capacity of konjac glucomannan may slow gastric emptying and small bower transit, thus increase satiety [29]. Those characteristics of konjac glucomannan are believed to be similar to those of other soluble dietary fibers, thus it can reduce glucose level due to the increased viscosity of gastrointestinal content, slow gastric emptying, and act as a barrier to mucosal diffusion [30]. The reduction effect of glucose was also considered temporary, which is confirmed by the result of the satellite group that did not show a significant change, compared to the control group. Furthermore, no morphological or pathological changes were observed in all vital organs, namely, the kidney, liver, heart, spleen, and lung, in the histopathological examination between the control and treated groups, indicating that the film formulation administration is nontoxic.

## 5. Conclusions

Based on the results of the current study, we may conclude that the film formulation that combines κ-carrageenan and konjac glucomannan was well tolerated up to a 75 mg/kg bw dose when orally administered to Wistar rats. The glucose, cholesterol and triglyceride changes in the treatment groups were considered to be temporary effects of the film suspension administration, because a normal result was achieved after two weeks of discontinuation of the test substance. Cumulatively, these data suggest that the film formulation poses a low potential health risk but further study on its chronic toxicity should be undertaken to seek the effect of chronic exposure to film formulation.

**Author Contributions:** I.K.A., S.N.S., and L.D.N.S. conceived and designed the experiments; N.N.W.S. performed the experiments; N.N.W.S., I.K.A., and L.D.N.S. analyzed the data; N.N.W.S. wrote the paper; and I.K.A., and L.D.N.S. reviewed manuscript.

**Funding:** This research was funded by the Higher Education DP2M through the Decentralized Research scheme (Universities Excellent Research) ITB 2016.

**Conflicts of Interest:** The authors declare no conflict of interest.

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
