# Peer review of "Acute and Subchronic (28-day) Oral Toxicity Studies on the Film Formulation of k-Carrageenan and Konjac Glucomannan for Soft Capsule Application"

_scipharm, doi:10.3390/scipharm87020009_

Reviewer 1 Report

The manuscript presents the toxicity evaluation of a film formulation made of k-Carrageenan and Konjac for future for soft capsule application. The article is well written and the methods used seem to be correct. There are some typing mistakes:

It should be methylparaben instead of methyl paraben

It should be Wistar instead of wistar

Other remarks:

At row 75: Normal female wistar rats were used for acute toxicity study. Normal male and female wistar
76 rats were used for the subchronic (28-day) study. Please provide a reference for using only females for the acute toxicity.

Author Response

There are some typing mistakes:

It should be methylparaben instead of methyl paraben

Response 1: Correction was made

It should be Wistar instead of wistar

Response 2: Correction was made

Other remarks:

At row 75: Normal female wistar rats were used for acute toxicity study. Normal male and female wistar
76 rats were used for the subchronic (28-day) study. Please provide a reference for using only females for the acute toxicity.

Response 3: Only female Wistar rats was used for the acute toxicity refer to OECD 420 Guideline, 2001 for Acute Oral Toxicity – Fixed Dose Procedure. The guidance on selection of animal species (point 9) states that the preferred rodent species is the rat, although other rodent species may be used. Normally females are used. This is because literatures surveys of conventional LD50 tests show that usually there is little difference in sensitivity between the sexes, but in those cases where differences are observed, females are generally slightly more sensitive.

Reviewer 2 Report

The paper fits the scope of Sci. Pharm. journal and the authors present the evaluation of the acute and sub-chronic oral toxicity of a pharmaceutical film containing two well-known additives. The research protocols are appropriate and the work is well structured. However, in order to improve the manuscript, I would like to make some remarks.

Introduction

row 37 amorphophallus konjac should be written Amorphophallus konjac K. Koch

rows 43-44. The reference is missing.

Materials and methods

row 75 Please correct wistar (Wistar)

row 78 ad libitum should be italic

2.4.3. section - reference is missing

Statistical software should be indicated.

Staining method reference is missing.

Please be more specific and indicate the local animal care committee.

Results

3.1 Acute oral toxicity study

Why no necropsy and histopathology examination was performed on the animals after the 14 days period of testing?

Discussion

An analysis of the quantity of k-Carrageenan and Konjac Glucomannan should be provided. The actual quantities of the two components are low and variable (due to the variability of the parameter loss on drying). Thus, a discussion on the two additive administrated doses should be made so that the author's work to be useful in further studies.

The swelling capability of glucomannan at the lower dose does not explain the glucose level. As stated by the authors, at doses higher than 10mg/Kg bw of film the glucomannan should not have swelling capability, fact which it's not true. Please revise this discussion.

Author Response

Introduction

row 37 amorphophallus konjac should be written Amorphophallus konjac K. Koch

Response 1: Correction was made

rows 43-44. The reference is missing.

Response 2: The reference for those sentences refer to the reference states on row 47 : [5] Meeting Joint FAO/WHO Expert Committee on Food Additives, Organization WH: Compendium of Food Additive Specifications: Addendum 9. Food & Agriculture Org; 2001.

Materials and methods

row 75 Please correct wistar (Wistar)

Response 3: Correction was made

row 78 ad libitum should be italic

Response 4: Correction was made

2.4.3. section - reference is missing

Response 5: The reference for parameters tested on hematological analysis was added: [8] OECD Guidelines for The Testing of Chemicals: Repeated Dose 28-Day Oral Toxicity Study in Rodents, OECD/OCDE 407, Adopted: 3 October 2008

Statistical software should be indicated.

Response 6: Statistical software was added

Staining method reference is missing.

Response 7: The reference for staining method was added

Please be more specific and indicate the local animal care committee.

Response 8: Local animal care committee is The Health Research Ethic Committee of Faculty of Medicine Universitas Padjajaran Bandung

Results

3.1 Acute oral toxicity study

Why no necropsy and histopathology examination was performed on the animals after the 14 days period of testing?

Response 9: During acute oral toxicity study, no signs of toxicity nor death were observed. After 14 day of treatment, gross necropsy was performed for all tested animal and macroscopic observation on organs namely heart, liver, spleen, lung, and kidney were done. The gross necropsy results no abnormality of the organs and considered no damage on the organs thus the histopathological study was not performed. This acute toxicity study showed that the film formulation of k-carrageenan and konjac glucomannan was not toxic after an acute exposure.

Discussion

An analysis of the quantity of k-Carrageenan and Konjac Glucomannan should be provided. The actual quantities of the two components are low and variable (due to the variability of the parameter loss on drying). Thus, a discussion on the two additive administrated doses should be made so that the author's work to be useful in further studies.

Response 10: The discussion of the quantity of polymers was added. Low polymer content was applied due to limitation of high viscosity produced by the polymers but 5% total polymer content able to obtain elastic film that suitable for soft capsule application. The film was produced from interaction of two polymers through gelling process that involved double helix conformation and ionic bonding.  The doses of test substance for toxicity studies were confirmed by prepare the film suspension using calculated anhydrate film weight.

The swelling capability of glucomannan at the lower dose does not explain the glucose level. As stated by the authors, at doses higher than 10mg/Kg bw of film the glucomannan should not have swelling capability, fact which it's not true. Please revise this discussion.

Response 11: Konjac Glucomannan has ability to retain water with high absorptive properties and swell to roughly 200 times their original volume. That properties cause low doses of glucomannan able to prolong gastric emptying time which increases satiety.

The statement ‘the reduction of the glucose level was related to the swelling capability of glucomannan’ was deleted. The discussion revised to “The high water-binding capacity and swelling capacity of konjac glucomannan may slow gastric emptying and small bower transit thus increased satiety. Those characteristics of konjac glucomannan is believed to be similar to those of other soluble dietary fibers thus can reduce glucose level due to it increase viscosity of gastrointestinal contents, slows gastric emptying, and acts as a barrier to mucosal diffusion.”

References:

- Chengquan Tan, Hongkui Wei, Xichen Zhao, Chuanhuai Xu, Yuanfei Zhou, and Juan Peng. Soluble fiber with hing water-binding capacity, swelling capacity, and fermentability reduces food intake by promoting satiety rather than satiation in rats, Nutriens, 2016, 8, 1 – 15

- Supornpim Chearskul, Somkiat Sangurai, Wannee Nitiyanant, Wantanee Kriengsinyos, Suwattanee Kooptiwut, and Tasma Harindhanavudhi. Glycemic and lipid responses to glucomannan in Thais with type 2 diabetes mellitus, J Med Assoc Thai, 2007, 90(10), 2150-2156

Round  2

Reviewer 2 Report

The authors made all the corrections to the paper. The improved version of the manuscript it's suitable for publication in Sci. Pharm.